# Probing the topologically trivial nature of end states in antiferromagnetic atomic chains on superconductors

Lucas Schneider [1], Philip Beck [1], Levente Rózsa [2,3,4], Thore Posske [5,6], Jens Wiebe [1] ✉ & Roland Wiesendanger [1]

Spin chains proximitized by *s*-wave superconductors are predicted to enter a mini-gapped phase with topologically protected Majorana modes (MMs) localized at their ends. However, the presence of non-topological end states mimicking MM properties can hinder their unambiguous observation. Here, we report on a direct method to exclude the non-local nature of end states via scanning tunneling spectroscopy by introducing a locally perturbing defect on one of the chain's ends. We apply this method to particular end states observed in antiferromagnetic spin chains within a large minigap, thereby proving their topologically trivial character. A minimal model shows that, while wide trivial minigaps hosting end states are easily achieved in antiferromagnetic spin chains, unrealistically large spin-orbit coupling is required to drive the system into a topologically gapped phase with MMs. The methodology of perturbing candidate topological edge modes in future experiments is a powerful tool to probe their stability against local disorder.

Hybrid systems of magnetic and superconducting materials in reduced dimensions have been of great interest in recent years, owing to the exciting emergent physics such as unconventional superconductivity and topological edge modes expected in these platforms[1–6]. In particular, there has recently been a focus on antiferromagnetic materials proximity coupled to *s*-wave superconductors[7–11]. Since antiferromagnets possess no net magnetic moment in their magnetic unit cell, they do not have considerable stray fields which would destroy superconducting order. A coexistence of antiferromagnetism and superconductivity in, e.g., thin films on *s*-wave superconductors[8] or the Fe-based superconductors[12,13] was previously explained by the large size of Cooper pairs compared to the magnetic unit cell of the materials or by an unconventional $s_{\pm}$ type pairing symmetry[14]. The absence of the net magnetic moment also gives rise to an effective

time-reversal symmetry (ETRS) in an antiferromagnet, consisting of physical time reversal inverting the spin directions and a spatial symmetry exchanging the antiferromagnetic sublattices.

In the limit of a single magnetic adatom or molecule on a superconducting surface, the interaction of its spin with the host material induces local quasiparticle states inside the superconducting gap known as Yu-Shiba-Rusinov (YSR) states[15–17]. When multiple of these impurities are close to each other, their YSR states split in energy as they start to couple[18,19]. Without spin-orbit coupling (SOC), such a splitting would be prohibited by the ETRS for a strictly antiferromagnetic alignment of the spins in adatom pairs[18,20]. However, it has been shown recently that the splitting is allowed in the presence of SOC on a surface[19,21]. In larger arrays of magnetic impurities, the coupled YSR states form YSR sub-gap bands, which can potentially have

[1]Department of Physics, University of Hamburg, D-20355 Hamburg, Germany. [2]Department of Physics, University of Konstanz, D-78457 Konstanz, Germany. [3]Department of Theoretical Solid State Physics, Institute of Solid State Physics and Optics, Wigner Research Centre for Physics, H-1525 Budapest, Hungary. [4]Department of Theoretical Physics, Budapest University of Technology and Economics, H-1111 Budapest, Hungary. [5]I. Institute for Theoretical Physics, University of Hamburg, D-22607 Hamburg, Germany. [6]Centre for Ultrafast Imaging, Luruper Chaussee 149, D-22761 Hamburg, Germany. ✉e-mail: jwiebe@physnet.uni-hamburg.de

non-trivial topology[3–6,22,23] and lead to the emergence of topologically protected Majorana modes (MMs) at the edges of the array[24–28]. In one dimension, a chain of magnetic impurities with topologically non-trivial YSR bands is expected to host zero-energy MMs at both ends for sufficient chain length. The Majorana number in spin chains can be interpreted as the parity of the number of spin-polarized bands crossing the Fermi level in the absence of superconducting pairing terms (c.f. Methods, Eq. (8)). Without SOC, the ETRS in anti-ferromagnetic chains leads to doubly degenerate excitations in the magnetic Brillouin zone. Thus, there is necessarily an even number of band crossings and a topologically trivial Majorana number. However, finite SOC breaks this symmetry and the degeneracies can be lifted. Thus, SOC or certain spatial symmetries theoretically open up possibilities for topologically non-trivial phases hosting MMs also in anti-ferromagnetic chains[9,10]. Experimental investigations of the low-energy electronic structure with a focus on such modes so far largely concentrated on ferromagnetic chains[25,27,29–35] and a few on spin-spirals[24,36], but studies of the antiferromagnetic case are sparse[34].

A general problem with the interpretation of such experimental data is the fact that near-zero-energy states can always appear as artifacts—e.g., induced by local defects or by a different electronic structure at the chain termination—in local tunneling spectroscopy measurements[33,34,36]. Therefore, a good understanding of the sample's underlying YSR band structure and its direct correlation with the observation of end states is clearly desired to pin down the nature of these end states. In this work, we additionally pursue a new strategy to test the MM nature of end states residing in a comparably large bulk minigap which we observed for scanning-tunnel-microscope-(STM)-tip-constructed[37] antiferromagnetic Mn chains on the atomically clean surfaces of Nb(110) and Ta(110): We intentionally locally perturb one

end of the chain with a local defect. While MMs residing in a topological minigap which is wider in energy than the perturbation's energy scale are expected to remain either completely unaffected or will merely laterally shift, trivial end states will split in energy only on the perturbed side of the chain (see Supplementary Note 1). We compare the experimental findings to an effective single-particle model for an antiferromagnetic spin chain coupled to an s-wave superconductor.

## Results

### Antiferromagnetic Mn chains on Nb(110)

Single Mn atoms on clean Nb(110) and Ta(110) surfaces have been studied both experimentally[19,27,29,32,38,39] and theoretically[40–43], and offer a suitable platform for studying well-defined YSR arrays. In particular, it has been shown that the magnetic interaction between neighboring adatoms can be tuned from ferromagnetic to antiferromagnetic when varying the inter-atomic distance and the crystallographic direction connecting the atoms on the surface[19,38]. Spin-polarized measurements have revealed that densely packed linear chains of Mn atoms constructed along the [1$\bar{1}$1] direction of the (110) surfaces (Fig. 1a) of Nb[38] and Ta (Supplementary Fig. 2) feature an out-of-plane anti-ferromagnetic ground state, in agreement with ab-initio calculations[42].

We start by presenting the results on antiferromagnetic Mn chains on Nb(110): Fig. 1b shows the topography of a $Mn_{40}$ chain together with examples of deconvoluted d$I$/d$V$ maps obtained at sub-gap energies ($\Delta_{Nb}$ = 1.51 meV, see Methods, Supplementary Note 3 and Supplementary Fig. 3). A d$I$/d$V$ line profile along the same chain is presented in Fig. 1d. Additionally, the evolution of the sub-gap local density of states (LDOS) for $Mn_N$ chains on Nb(110) with increasing number of sites $N$ is shown in Fig. 1c, separately for the chain's left end (left panel), for the chain's bulk (central panel) and for the right end (right panel).

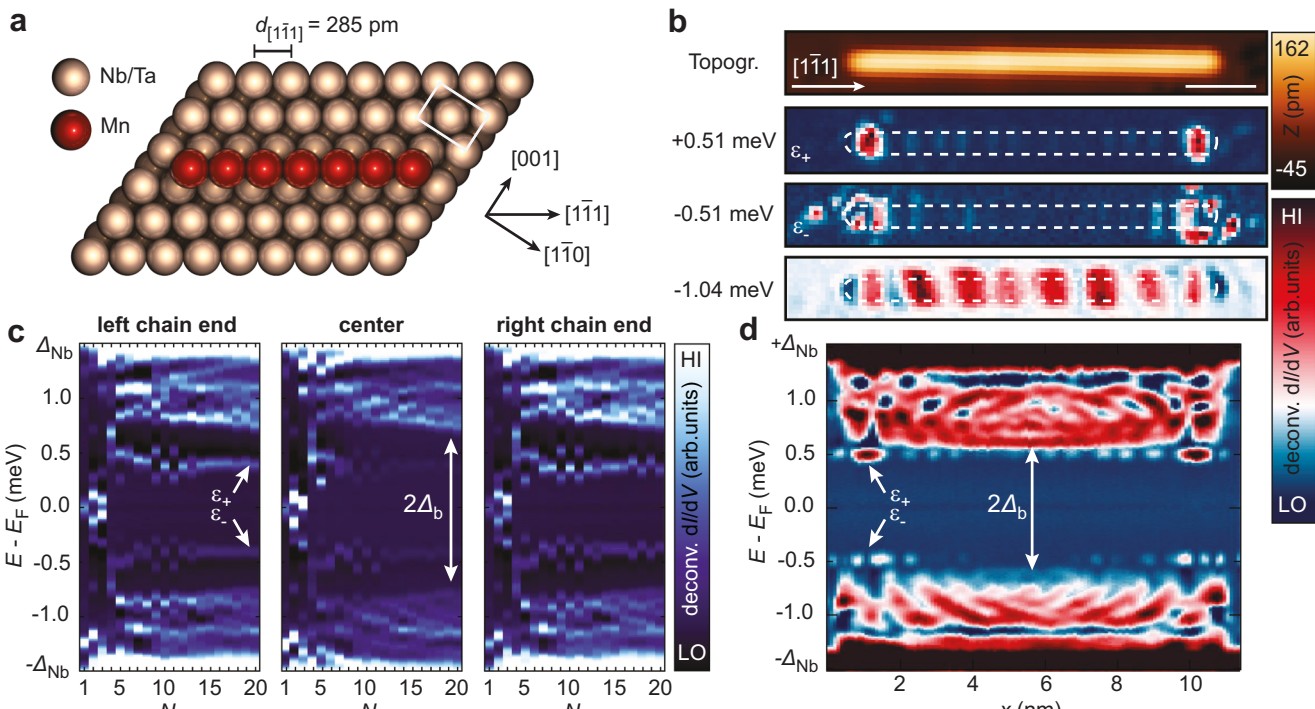

**Fig. 1 | Geometry and low-energy electronic structure of antiferromagnetic Mn chains on Nb(110). a** Sketch of Mn adatoms (red) positioned on neighboring hollow sites of the Nb(110) or Ta(110) lattice (beige) along the [1$\bar{1}$1] direction. **b** Constant-current STM image (topography, top panel) of a $Mn_{40}$ chain and simultaneously acquired deconvoluted d$I$/d$V$ maps (bottom panels) at selected energies as indicated. The white scale bar corresponds to 2 nm. The apparent extent of the chain from the top panel is marked by white dashed boundaries. **c** Sequence of deconvoluted d$I$/d$V$ spectra measured on the left end, in the center and on the right end of $Mn_N$ chains with increasing number of sites $N$. The emergence of the chain's bulk minigap $\Delta_b$ and of finite-energy end states $\varepsilon_{+/-}$ are indicated. Note that single atoms are added only to the right chain end during this measurement. **d** Deconvoluted d$I$/d$V$ line-profile measured along the longitudinal axis through the center of the $Mn_{40}$ chain. The lateral position of the spectra is aligned with the topography in panel b. Parameters: $V_{stab}$ = −6 mV, $I_{stab}$ = 1 nA, $V_{mod}$ = 20 μV.

There is a continuum of states visible in the energy range of 0.7 meV < |E| < 1.5 meV, exhibiting standing-wave patterns (c.f. Fig. 1d and the map at −1.04 meV in Fig. 1b) due to quasiparticle interference (QPI). This indicates the formation of dispersive YSR bands[27,32,34] which we analyze later on. In contrast to this, no bulk states are observed within a gapped region of $\pm\Delta_b = \pm 0.7$ meV for chain lengths exceeding $N = 6$ sites. This is a surprising result since the maximal gaps previously found in dispersive YSR bands in ferromagnetic spin chains on superconducting surfaces were on the order of 50-180 μeV[24,27,30,32,44].

Inside the minigap $\Delta_b$, we find clearly localized end states at energies $\varepsilon_{+/-} = \pm 0.51$ meV (Fig. 1b). In addition to the end-state nature of these features, a small oscillatory component of their wave function is found to decay into the chain's bulk (see Fig. 1b, d). As it can be seen in Fig. 1c, these energetically isolated states form at energies $\varepsilon_{+/-}$ already for $N > 5$ and their energy is only faintly oscillating in energy for longer systems. This fast convergence of the end-state energy with increasing chain length agrees with the good localization of the features, i.e., interactions between the ends already vanish for short chains. Notably, the end states in the regime $8 < N < 14$ are energetically split into four eigenstates for odd $N$ while there are only two eigenstates for even $N$ (see Fig. 1c).

## Comparison to antiferromagnetic Mn chains on Ta(110)

In order to further investigate experimentally whether the appearance of such low-energy end states is a consequence of the antiferromagnetic spin alignment in the chain and thus not limited to only one experimental platform, we study structurally similar Mn chains on a clean Ta(110) surface ($\Delta_{Ta} = 0.64$ meV). It has been shown previously that Mn atoms on a Ta(110) surface exhibit surprising similarities to Mn/Nb(110)[29,39] due to the identical number of valence electrons and the same crystal structure of the substrates, however, with a strongly enhanced SOC in Ta compared to Nb. Most notably, densely packed Mn chains along the [1̄1̄1] direction are also found to be antiferromagnetically ordered (see Supplementary Fig. 2). Measurements of the low-energy electronic structure in these chains are presented in Fig. 2. The d$I$/d$V$ maps in Fig. 2a measured around a Mn$_{22}$ chain reveal the presence of well-localized end states with near-zero energy and a similar spatial appearance as the end states found in Mn chains on Nb(110), while higher-energy excitations are mostly localized in the chain's bulk. Furthermore, a weak oscillatory pattern can be seen in the map obtained at −0.28 meV, indicating the presence of QPI in the bulk states of this platform as well. The d$I$/d$V$ line-profile along a Mn$_{20}$ chain shown in Fig. 2b suggests that the bulk of the chain is electronically gapped by a minigap $\pm\Delta_b = \pm 0.2$ meV. Figure 2c shows the length dependence of the LDOS features for chains with $10 < N < 22$. Here, it is also visible that end states of almost constant energy are present on both chain ends while the bulk remains gapped. On the left, unperturbed end, a damped even-odd oscillation in the end-state energies is observed again, as for the Nb case. The overall appearance of the low-energy electronic structure is very similar to Mn/Nb(110). It is therefore natural to conjecture that the end states in the Mn/Ta(110) platform have a similar origin as in the Mn/Nb(110) case. However, since the end states in Mn/Ta(110) are very close to or at zero energy, the question arises whether or not this system realizes a topological superconductor with the accompanying near-zero energy MMs at its ends.

## Perturbation of the end states by local defects

For testing the topologically non-trivial or trivial nature of the end states, the influence of local defects on the end states in Mn/Nb(110) and Mn/Ta(110) chains is studied. These defects can be either of magnetic or of non-magnetic origin. It has been shown previously that the energy of individual YSR states is very sensitive to

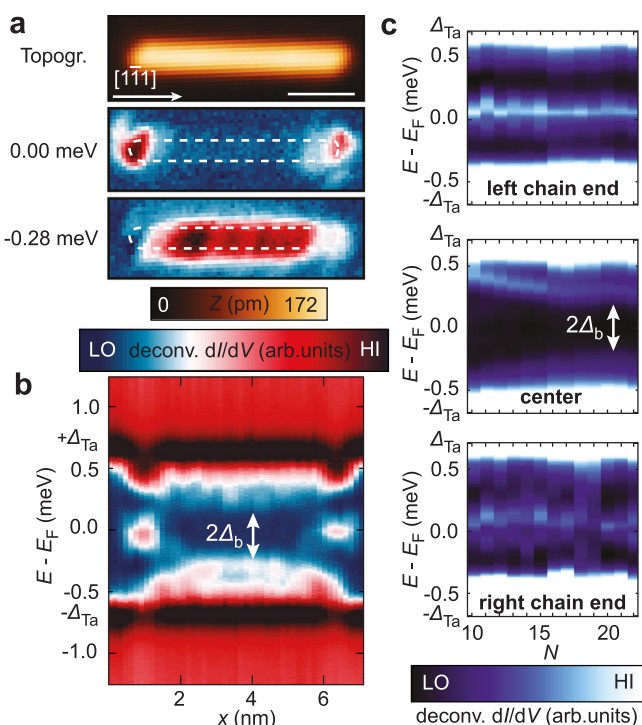

**Fig. 2 | Sub-gap electronic structure of antiferromagnetic Mn chains on Ta(110). a** Constant-current STM image (topography, top panel) of a Mn$_{22}$ chain and simultaneously acquired deconvoluted d$I$/d$V$ maps (bottom panels) at selected energies as indicated. The white scale bar corresponds to 2 nm. The apparent extent of the chain from the top panel is marked by white dashed boundaries. **b** Deconvoluted d$I$/d$V$ line-profile measured along the longitudinal axis through the center of a Mn$_{20}$ chain. The lateral position of the spectra is aligned with the topography in **a**. The arrow indicates the chain's bulk minigap. **c** Sequence of deconvoluted d$I$/d$V$ spectra measured at the left end, in the center and at the right end of Mn$_N$ chains with increasing number of sites $N$. The emergence of a bulk minigap $\Delta_b$ (marked) and of end states with energies close to the Fermi energy $E_F$ can be observed. Parameters: $V_{stab} = -2.5$ mV, $I_{stab} = 1$ nA, $V_{mod} = 20$ μV.

variations in adsorption geometries[45,46], defects like local oxygen impurities[47], or local charge density[48]. Also hydrogenation of adatoms has been shown to drastically alter their magnetic properties and, importantly, their exchange coupling to the substrate[49]. Since the exchange coupling strength is one of the main factors determining the YSR state energies of magnetic impurities[17], hydrogenated Mn atoms at the end of the chain are expected to have clearly shifted YSR state energies compared to unperturbed Mn atoms. Figure 3a shows an example of a Mn$_{20}$ chain on Nb(110) with a dark spot visible on the left side of the chain, which is presumably adsorbed hydrogen or another weakly bound surface adsorbate trapped at an oxygen defect of the Nb(110) surface. When measuring d$I$/d$V$ maps at sub-gap energies on this Mn chain, the left and the right end state have slightly different energies (±0.43 meV and ±0.35 meV, respectively, see Supplementary Note 4 for details). When removing the defect next to the chain by local voltage pulses (Fig. 3b), both end states appear at the same energy (±0.35 meV) again. A similar effect can actually be seen in the data of Fig. 1c already where the end state on the right end is found to oscillate in energy for $15 < N < 20$ while the state on the left end remains at fixed energy. This unambiguously proves the local nature of these states, in clear contrast to non-local, spatially correlated states like MMs or their precursors[27] (see Supplementary Note 1 for the effect of potential disorder within a minimal model for antiferromagnetic YSR chains). Note at this point that although the finite energy of the

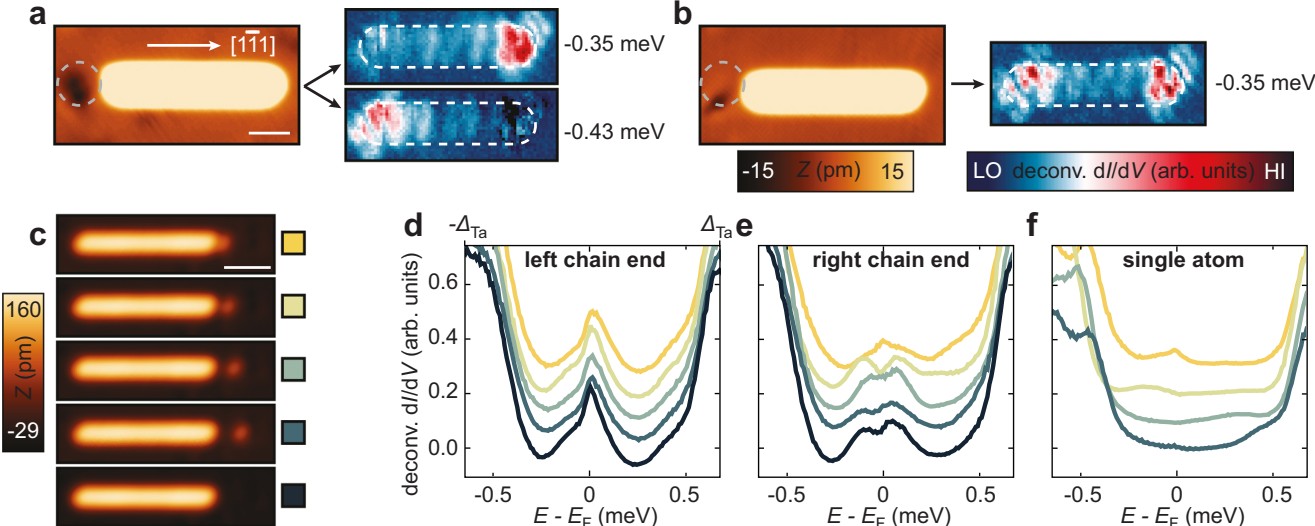

**Fig. 3 | Response of the end state energies to local defects. a** Constant-current image of a Mn$_{20}$ chain on Nb(110) (left panel) with adjusted contrast to highlight the defect on the left side of the image which is marked by the gray dashed circle. The white scale bar corresponds to 1 nm. Deconvoluted d$I$/d$V$ maps at selected energies (right panels) reveal that the two end states have different energies. The apparent extent of the chain from the left panel is marked by white dashed boundaries. **b** Constant-current image of the same Mn$_{20}$ chain after applying bias voltage pulses to remove the local defect. A deconvoluted d$I$/d$V$ map at a selected energy (right) shows that the end states now have equal energy. Parameters: $V_{stab} = -6$ mV,

$I_{stab} = 1$ nA, $V_{mod} = 20$ μV. **c** Constant-current STM images of a Mn$_{22}$ chain on Ta(110) and a single Mn atom which is subsequently moved to positions with different distances from the right chain end. The bottommost panel shows the unperturbed chain. The white bar corresponds to 2 nm. Parameters: $V_{stab} = -20$ mV, $I_{stab} = 0.2$ nA. **d** Deconvoluted d$I$/d$V$ spectra measured on the left chain end, **e**, on the right chain end and **f**, on the single Mn atom for the different adatom distances depicted in **c**. The spectra are vertically offset for clarity. Their color indicates which panel in **c** they belong to. Parameters: $V_{stab} = -2.5$ mV, $I_{stab} = 1$ nA, $V_{mod} = 20$ μV.

end states already suggested their topologically trivial origin, it has been shown previously that topological states may remain at finite energies even for long chains due to fine-tuned interactions[50].

We performed a similar experiment for Mn chains on Ta(110) using a single Mn atom as the local defect. Figure 3c shows topographies of a Mn$_{22}$ chain with an additional Mn atom, whose distance to the right chain end is varied. d$I$/d$V$ spectroscopy on the left and right chain end as well as on the single atom reveals the coupling between both structures (Fig. 3d–f): as the single atom approaches the chain, its sub-gap spectral character is altered. The same holds for the right chain end, which directly interacts with the single atom because of their close proximity, i.e. the peak position of the end state slightly shifts. In contrast, the left chain end is not altered, proving that the end states are entirely local instead of collective properties of the chain. This is not expected for MMs or their precursors, where moving a magnetic atom close to one chain end merely laterally shifts the spatial location of the zero-energy end state on that side, or shifts the energy of the lowest-lying state simultaneously at both ends of the chain[27]. Additional data provided in Supplementary Note 5 demonstrates that this is not the case, i.e. the spectral weight is not just laterally shifted. The observed energetical splitting of the two end states with respect to each other indicates that in the undisturbed chains, two states are localized, one at each end, which are degenerate due to the antiferromagnetic spin structure, in contrast to MMs or their precursors that form a single fermionic state with enhanced intensities at both ends.

### Minimal model for antiferromagnetic YSR chains

To understand the nature of these trivial end states, we construct a minimal theoretical model for antiferromagnetic YSR chains. A single-particle model following refs. 3,4,6,22,23 successfully describes the sub-gap electronic bands in ferromagnetic YSR chains, especially when extended with local potential scattering[27,32]. We extend these models by studying an antiferromagnetic chain on a superconducting substrate including Rashba SOC and arrive at the following minimal

next-nearest-neighbor model Hamiltonian (see Methods for details):

$$
\begin{aligned}
\mathcal{H} = &-E_0 \sum_{i=1}^{N} c_i^\dagger c_i - t_1 \sum_{i=1}^{N-1} (c_i^\dagger c_{i+1} + \text{h.c.}) - t_2 \sum_{i=1}^{N-2} (c_i^\dagger c_{i+2} + \text{h.c.}) \\
&- \Delta_1 \sum_{i=1}^{N-1} (c_i^\dagger c_{i+1}^\dagger + \text{h.c.}) - \Delta_2 \sum_{i=1}^{N-2} (c_i^\dagger c_{i+2}^\dagger + \text{h.c.}).
\end{aligned}
\tag{1}
$$

Here, $c_i^\dagger$, $c_i$ represent the creation and annihilation operators of YSR states at site $i$ of a one-dimensional chain with $N$ sites. The on-site energies $-E_0$ would correspond to the YSR state energies of the individual Mn impurity. The model includes nearest-neighbor (NN: index 1) and next-nearest-neighbor (NNN: index 2) hopping ($t_1$, $t_2$) and superconducting pairing ($\Delta_1$, $\Delta_2$) (Fig. 4a). The model connects to the Kitaev chain[51] if only the NN terms are kept.

Notably, a perfectly collinear antiferromagnetic ordering is characterized by an ETRS (see Methods) consisting of physical time reversal inverting the spin directions and a translation by the distance between the chain atoms[10]. In the present Hamiltonian expressed in the basis of atomic YSR states, the ETRS implies that hopping terms are only allowed between atomic YSR states with the same spin, i.e. between NNNs in the present model (see Methods and Fig. 4a). In contrast, the effective superconducting pairing $\Delta_1$ can easily be induced between adjacent atoms, since their spins are anti-aligned, and will be suppressed for NNNs ($\Delta_2$), where the spins have a parallel alignment. For non-zero SOC these restrictions are lifted (see Methods), and the ratio of the coefficients induced by the SOC and present without the SOC may be estimated by the dimensionless parameter $\alpha_R/\hbar v_F$, where $\alpha_R$ is the Rashba parameter and $v_F$ is the Fermi velocity. This parameter was estimated to be $k_h/k_{F,0} = 0.094$ for Mn/Nb(110) in ref. 27. We consequently pay particular attention to the case $t_1 \ll t_2$, and $\Delta_1 \gg \Delta_2$ of our minimal model and further assume $E_0 \approx 0.0$ meV, which is motivated by the experimental YSR state energies for $N = 1,2,3$ (Fig. 1c).

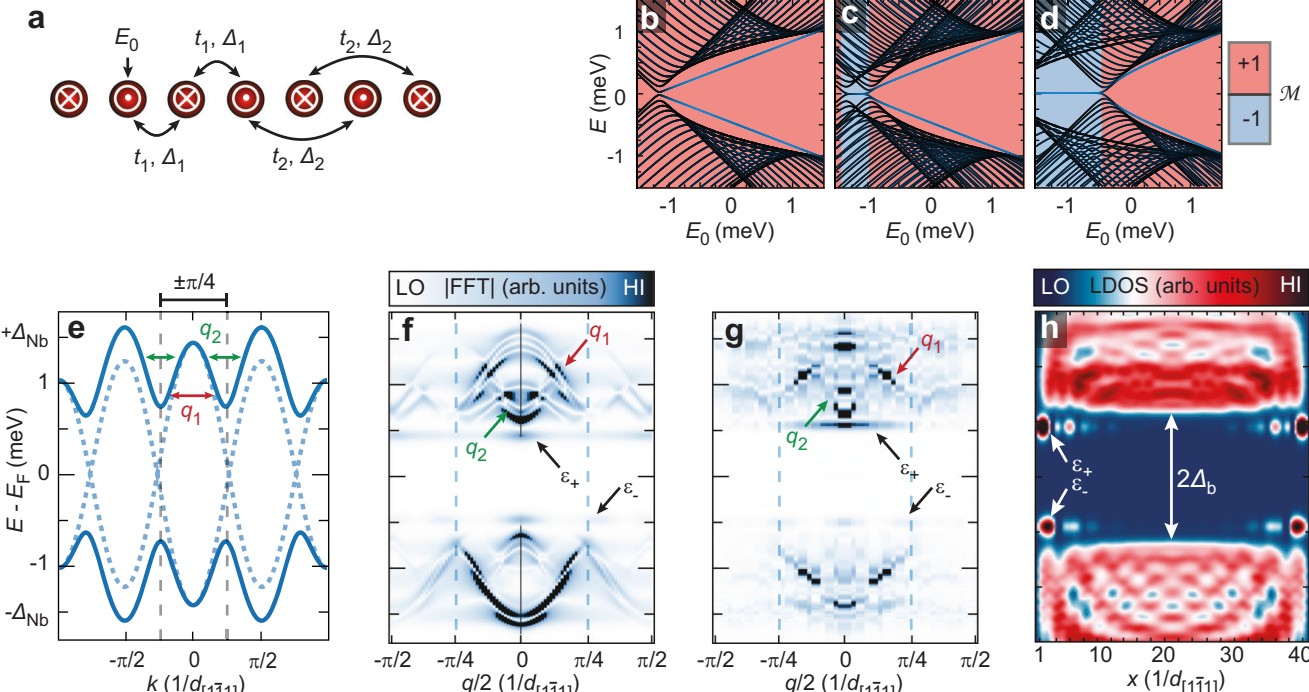

**Fig. 4 | Minimal model for antiferromagnetic YSR chains and experimental QPI results. a** Sketch of the parameters in an antiferromagnetically ordered chain considered in the minimal model of the YSR states. The symbols indicate up ($\odot$) and down ($\otimes$) oriented magnetic moments, respectively. $t_1$ ($t_2$) is the NN (NNN) hopping, $\Delta_1$ ($\Delta_2$) is the NN (NNN) pairing term and $E_0$ is the on-site energy (see Methods). **b** Eigenenergies of a finite-size chain with $N=40$ for varying on-site energy $E_0$ and hopping parameters $t_1 = 0.0$ meV, $t_2 = 0.6$ meV. **c** Same as panel b, but with a small ETRS breaking term, $t_1 = 0.1$ meV, $t_2 = 0.6$ meV. **d** Same as **b** and **c**, but with an unrealistically large ETRS breaking term, $t_1 = 0.4$ meV, $t_2 = 0.6$ meV. $\Delta_1 = 0.5$ meV, $\Delta_2 = 0.0$ meV are chosen for all **b**–**d**. The Majorana number $\mathcal{M}$ of the system is trivial ($+1$) in the red regions and non-trivial ($-1$) in the blue regions (see

Methods for details). The dark blue lines in panels b–d are end states of the system. **e** YSR band structure from the minimal model using the parameters $E_0 = 0.0$ meV, $t_1 = 0.1$ meV, $t_2 = 0.6$ meV, $\Delta_1 = 0.0$ meV (dashed lines) as well as $\Delta_1 = 0.5$ meV (solid lines), and $\Delta_2 = 0.0$ meV. The position of $k = \pm\pi/4$ is marked by dashed lines. **f** Absolute values of the line-wise FFT of a calculated LDOS line-profile similar to panel h using $N = 100$ sites. **g** Absolute values of the line-wise FFT of the experimental d$I$/d$V$ line-profile in Fig. 1d. The supposed dispersive scattering vectors $q_1, q_2$ and the edge state energies $\varepsilon_{+,-}$ are marked in **e**–**g**. **h** Calculated LDOS along a chain of $N = 40$ sites using the parameters from panels e and f. The energies of the finite-energy end states $\varepsilon_{+/-}$ and the chain's bulk minigap $\Delta_b$ are marked.

Experimentally, information about the sub-gap bands' dispersion can be extracted from one-dimensional quasiparticle interference (QPI) measurements[27,32] (see Supplementary Note 6 and the extracted dispersion of the scattering vectors in Fig. 4g). We adjust the parameters of the minimal model such that the dispersions of the possible scattering vectors $q_1$, $q_2$ in Fig. 4f extracted from the calculated dispersion in Fig. 4e reasonably fit the experimental dispersions of Fig. 4g. Using these parameters, we calculate the spatially resolved LDOS for various chain lengths $N$ (Fig. 4h, Supplementary Fig. 6, and Methods) which, given the few free parameters, agree surprisingly well with the experimental data in Fig. 1c, d. In particular, the minimal model nicely reproduces the appearance of non-zero energy end states with an oscillatory decay towards the chain center at energies $\varepsilon_{+/-}$ in a large minigap $\Delta_b$, the fast convergence of these end states towards $\varepsilon_{+/-}$ already for short chains (Supplementary Fig. 6), as well as the dispersive nature of the bulk states.

Having substantiated the minimal model by comparison to the experimental data, we investigate its parameter-dependent phases and the topological nature of the end states in the following. Figure 4b–d show the energy spectrum of a 40-site chain vs. the on-site energy $E_0$ for different values of the hopping terms $t_1$ and $t_2$. It can be seen that the topologically non-trivial phase of the Kitaev chain is entirely quenched for $t_1 \to 0$ which corresponds to zero SOC (Fig. 4b). This is a consequence of the ETRS, which results in a Kramers degeneracy of all states, hence in a topologically trivial Majorana number. This finding agrees with previous studies, showing that topologically non-trivial phases cannot be found in antiferromagnetic YSR chains without

either strong SOC or additional supercurrents[9,10] breaking the ETRS. However, it is found that the end states at finite energy are formed even in this topologically trivial regime for $t_2 > t_1$ whenever the gap is reopened by the superconducting term $\Delta_1$. Each of these end states is twofold degenerate for even-length chains and shows a small splitting for odd-length chains where the ETRS is broken. This prediction agrees well with the experimentally observed energy splittings in the regime $8 < N < 14$ for odd $N$ in Fig. 1c. The degeneracy is restored for semi-infinite chains ($N \to \infty$). The fact that they merge with the continuum of states in Fig. 4b–d for large $E_0$ without a gap closure further proves their topologically trivial origin. For non-zero values of $t_1$, corresponding to finite SOC, the non-zero-energy end states in some part of the topologically non-trivial phase are preserved (Fig. 4c). But now, also a topologically non-trivial phase with zero-energy MMs is recovered, which only takes up a large proportion of the phase space for unrealistically large values of SOC (Fig. 4d).

In conclusion, we have shown that the in-gap quasiparticle structure of dense, antiferromagnetic YSR chains can be qualitatively described by a minimal model. In contrast to ferromagnetic chains, the antiferromagnetic structure facilitates the formation of a large minigap in the YSR band even without SOC. This observation may be qualitatively explained by the fact that all YSR states in a ferromagnetic chain possess the same spin polarization, meaning that the s-wave superconducting pairing in the substrate may only open a gap in the spectrum for non-zero SOC. For the antiferromagnetic chain the localized YSR states at the sites have an alternating spin polarization, which enables pairing between these states without SOC and can lead

to a considerably larger minigap. This large size of the minigap naturally leads to a strong localization of potential end states, since their extension is inversely proportional to the minigap width[27,52]. Therefore, antiferromagnetic chains appear to be more suitable for the realization of well-localized end states than their ferromagnetic counterparts. However, as visible from our simulations in Fig. 4b and as was shown in refs. 9 and 10, the formation of the topologically non-trivial phase now is a threshold effect: the SOC has to compete with the pairing potential, meaning that unrealistically large values of SOC have to be assumed in order to enable the formation of MMs. This is in contrast to the ferromagnetic chain, which is gapless in the absence of SOC and consequently an arbitrarily weak SOC may drive it into the topologically non-trivial regime. The most promising path towards combining the large minigap of antiferromagnetic chains with the easily achievable topologically non-trivial phases of ferromagnetic chains lies in the realization of YSR bands formed by non-collinear spin configurations[24,53].

Our minimal model furthermore reproduces the presence of intriguing finite-energy end states in the antiferromagnetic chains which are not topologically protected and whose energy can be tuned by local potentials. Notably, other experimental datasets on antiferromagnetic chains[34] show similar finite-energy end states, providing evidence that our findings are generally valid for other magnet-superconductor hybrids, too. A corresponding topologically trivial phase has been characterized by Pientka et al. as a two-channel p-wave superconducting wire where the interaction between two pairs of MMs lifts their energy to finite values and destroys their topological protection[3,54]. However, we want to emphasize that this separation into two channels is not straightforward if SOC is present, and thus, this argument must not be taken at face value. Coincidentally, the trivial end states may appear at near-zero energy by local potentials where they could be misinterpreted as MMs. We have shown here that a local perturbation of the end states with defects is a distinct way to prove their topologically trivial or non-trivial nature. This methodology can be used on other sample systems to probe the stability of candidate topological edge modes against local disorder.

## Methods
### Experimental procedures
The experiments were performed in a home-built STM setup under ultra-high-vacuum at a base temperature of $T = 320$ mK[55]. Nb(110) and Ta(110) single crystals were used as a substrate and cleaned by high-temperature flashes to $T > 2700$ K with an e-beam heater. In this way, atomically clean surfaces with only few residual oxygen impurities on the surface can be obtained for both materials, as shown previously[39,56]. Subsequently, single Mn atoms were deposited onto the surface of the clean substrates held at low temperatures ($T < 7$ K), resulting in a statistical distribution of adatoms. We use superconducting Nb tips made from mechanically cut and sharpened high-purity Nb wire. The tips were flashed in situ to about 1500 K to remove residual contaminants. STM images were obtained by regulating the tunneling current $I_{stab}$ to a constant value with a feedback loop while applying a constant bias voltage $V_{stab}$ across the tunneling junction. For measurements of differential tunneling conductance (d$I$/d$V$) spectra, the tip was stabilized at bias voltage $V_{stab}$ and current $I_{stab}$ as individually noted in the figure captions. In a next step, the feedback loop was switched off and the bias voltage was swept from -$V_{stab}$ to +$V_{stab}$. The d$I$/d$V$ signal was measured using standard lock-in techniques with a small modulation voltage $V_{mod}$ (RMS) of frequency $f = 4.142$ kHz added to $V_{stab}$. d$I$/d$V$ line-profiles and maps were acquired recording multiple d$I$/d$V$ spectra along a one-dimensional line or a two-dimensional grid of lateral positions on the sample, respectively. Note that we chose stabilization parameters at which the contribution of Andreev reflections and direct Cooper pair tunneling can be neglected (see Supplementary Note 3). The use of superconducting Nb tips increases the effective energy resolution of the experiment beyond the

Fermi-Dirac limit[57]. However, the differential tunneling conductance d$I$/d$V$ measured with superconducting tips is proportional to the convolution of the sample's local density of states (LDOS) and the superconducting tip density of states (DOS). Consequently, STS data need to be numerically deconvoluted in order to resemble the sample's LDOS, as it is typically known for the interpretation of STS experiments. After careful deconvolution of the spectra, the superconducting gaps of the Nb and Ta surfaces are found to be $\Delta_{Nb} = 1.51$ meV and $\Delta_{Ta} = 0.64$ meV, respectively (see refs. 27,32 for Nb and Supplementary Fig. 3 for Ta). We show only deconvoluted data throughout the manuscript (see Supplementary Note 3 for details). Mn chains were constructed using lateral atom manipulation[38,39] techniques at low tunneling resistances of $R \approx 30$–$60$ k$\Omega$.

### Minimal model for YSR bands in antiferromagnetic chains
Pientka et al. showed in ref. 3 that the low-energy electronic structure of a single-orbital chain of classical magnetic moments with a helical spin texture embedded in a three-dimensional superconducting host can be reduced to an effective Bogoliubov-de-Gennes Hamiltonian on a basis of projected YSR states. Subsequent models of the same type included Rashba-type SOC in ferromagnetic chains in refs. 2,4,6 and non-zero potential scattering and particle-hole asymmetric spectral weights in refs. 32,58. Here, we combine the SOC with non-zero potential scattering and a general spin structure, where the impurity atoms are assumed to be identical apart from the orientation of their classical spin. The effective Hamiltonian describing the YSR subgap band is given by

$$\mathcal{H} = \frac{1}{2} \sum_{i,j} \begin{bmatrix} \tilde{c}_i^\dagger & \tilde{c}_i \end{bmatrix} \begin{bmatrix} h_{ij} & \Delta_{ij} \\ -\Delta_{ij}^* & -h_{ij}^* \end{bmatrix} \begin{bmatrix} \tilde{c}_j \\ \tilde{c}_j^\dagger \end{bmatrix}, \quad (2)$$

with the matrix elements expressed as

$$h_{ij} = -E_0 \delta_{ij} + h_{ij}^{(0)} \langle \uparrow(i) | \uparrow(j) \rangle + h_{ij}^{(\alpha)} \langle \uparrow(i) | i\sigma^y | \uparrow(j) \rangle, \quad (3)$$

$$\Delta_{ij} = \Delta_{ij}^{(0)} \langle \uparrow(i) | \downarrow(j) \rangle + \Delta_{ij}^{(\alpha)} \langle \uparrow(i) | i\sigma^y | \downarrow(j) \rangle, \quad (4)$$

where $E_0$ is the single-impurity YSR energy. The real-valued coefficients $h_{ij}^{(0)}, h_{ij}^{(\alpha)}, \Delta_{ij}^{(0)}$ and $\Delta_{ij}^{(\alpha)}$ are material-specific constants, decaying with the distance $r = |\mathbf{r}_i - \mathbf{r}_j|$ as $\propto r^{-1} e^{ik_F r - r/\xi_0}$ for an isotropic electronic structure in three dimensions. Crucially, these terms do not depend on the magnetic structure of the chain but only on the electronic structure of the substrate and on the magnetic and non-magnetic scattering amplitudes of the impurities. Constants with superscript (0) remain finite at zero SOC, while the terms with superscript ($\alpha$) vanish. Exemplary formulae for these coefficients are given in refs. 4,6. We refrain from giving their full form here since they are not used explicitly in this manuscript, where the coefficients are fit to the experimentally observed YSR band structure, remaining consistent with the above form. Following a similar analysis to ref. 4, the most important conclusion is that the ($\alpha$) terms are linear in the dimensionless SOC parameter $\alpha_R/\hbar v_F$ in leading order, making them typically two orders of magnitude smaller than the (0) terms. Due to the oscillatory decay of the parameters with the distance, it might be possible but considerably difficult to design a system where the ($\alpha$) and (0) terms have comparable magnitude; decreasing the Fermi velocity in flat bands may also provide a way for achieving this.

The magnetic structure only enters in the matrix elements of the vectors $|\uparrow(i)\rangle = (e^{-i\varphi_i/2}\cos\vartheta_i/2, e^{i\varphi_i/2}\sin\vartheta_i/2)$ and $|\downarrow(i)\rangle = (e^{-i\varphi_i/2}\sin\vartheta_i/2, -e^{i\varphi_i/2}\cos\vartheta_i/2)$, which are eigenvectors of the spin operator $\mathbf{S}_i\boldsymbol{\sigma}$ with $\mathbf{S}_i = (\sin\vartheta_i\cos\varphi_i, \sin\vartheta_i\sin\varphi_i, \cos\vartheta_i)$ describing the magnetization direction of the $i$th impurity. In Eqs. (3) and (4), the Pauli matrix $\sigma^y$ enters due to the Rashba term when assuming that the chain is along the $x$ direction; for a different chain direction or symmetry

class, a different spin direction would be selected by the SOC. In the selected representation, the matrices possess the symmetry $h = h^\dagger$ and $\Delta = -\Delta^T$, and the particle-hole constraint may be represented in the usual form as $C = \tau^x K$, where $\tau^x$ exchanges the creation and annihilation operators and $K$ denotes complex conjugation.

We assume an antiferromagnetic spin structure with alternating sublattices $A$ and $B$ with $\vartheta_A = \pi - \vartheta_B = \vartheta$ and $\varphi_A = \varphi_B + \pi = \varphi$. This implies $\langle \uparrow(A)|\uparrow(B)\rangle = 0$ and $\langle \uparrow(A)|\downarrow(A)\rangle = 0$, i.e., the hopping $h_{ij}$ vanishes between sites at an odd distance and the pairing $\Delta_{ij}$ vanishes between sites at an even distance, as discussed in the main text. Due to the alternating structure, we restrict the Hamiltonian to NN and NNN terms, which we justify by the compact spatial structure of the YSR states of single Mn atoms on Nb(110) and on Ta(110) and their comparably weak coupling for interatomic distances above 1 nm[19,29,32,39]. The matrices read

$$
h = \begin{matrix} A \\ B \end{matrix} \begin{bmatrix} \ddots & & & & & & \\ \cdots & t_2 - it_2' s_\parallel & t_1 s_\perp & -E_0 & -t_1 s_\perp & t_2 + it_2' s_\parallel & \cdots \\ & \cdots & t_2 + it_2' s_\parallel & -t_1 s_\perp^* & -E_0 & t_1 s_\perp^* & t_2 - it_2' s_\parallel & \cdots \\ & & & & & & \ddots \end{bmatrix},
$$

(5)

$$
\Delta = \begin{matrix} A \\ B \end{matrix} \begin{bmatrix} \ddots & & & & & & \\ \cdots & -\Delta_2 s_\perp & -\Delta_1 + i\Delta_1' s_\parallel & 0 & -\Delta_1 - i\Delta_1' s_\parallel & \Delta_2 s_\perp & \cdots \\ & \cdots & -\Delta_2 s_\perp^* & \Delta_1 + i\Delta_1' s_\parallel & 0 & \Delta_1 - i\Delta_1' s_\parallel & \Delta_2 s_\perp^* & \cdots \\ & & & & & & \ddots \end{bmatrix}
$$

(6)

where $s_\parallel = \sin\vartheta\sin\varphi$ and $s_\perp = -\cos\varphi - i\cos\vartheta\sin\varphi$ are the components of the impurity spin parallel and perpendicular to the $y$ direction preferred by the SOC. The Hamiltonian of the antiferromagnetic chain possesses the ETRS $T_{\text{eff}} = U_B R_1 K$, where $R_1$ is translation by a lattice constant and $U_B$ adds a negative sign on sublattice $B$ and acts as identity on sublattice $A$.

For $s_\perp = 0$, the ETRS may be rewritten in Fourier space as a Kramers symmetry $T_{\text{eff}}^2 = -1$, enforcing the pairwise degeneracy of the states. If the impurity spins have a component perpendicular to the direction selected by the SOC, then one obtains $T_{\text{eff}}^2 \neq -1$ and the Kramers degeneracy or the ETRS is broken in this sense. This can be assumed to be the case here, since the spins have an out-of-plane component while the SOC selects an in-plane direction. If we set $s_\parallel = 0$ which does not influence whether ETRS is broken or not, and chose $s_\perp = 1$ to be real which can be achieved by an appropriate rotation of the spin quantization axes around $y$, then the local gauge transformation $c_j = (-1)^j U_B i\tilde{c}_j$ and $c_j^\dagger = (-1)^j U_B i\tilde{c}_j^\dagger$, where $(-1)^j U_B$ describes a sign change after every two lattice sites, transforms the Hamiltonian in Eqs. (2), (5) and (6) to the form given in Eq. (1) in the main text.

The following analysis is introduced in ref. 32. Here, we repeat the necessary details and adjust it to the antiferromagnetic system. Starting from Eq. (1) in the main text, the LDOS as a function of energy $E$ and position $x$ along a one-dimensional lattice of $N$ sites in Fig. 4h is computed by exact diagonalization of the low-energy Hamiltonian in Eq. (1) and summing over all pairs of eigenvalues $E_i$ and eigenvectors $\psi_i$:

$$
\text{LDOS}(E, x) = \sum_i \left[ P|\psi_{i,e}(x)|^2 + (1-P)|\psi_{i,h}(x)|^2 \right] \left( -\frac{\partial f(E - E_i, T = 320\,\text{mK})}{\partial E} \right)
$$

(7)

with the respective particle- (e) and hole-components (h) of the solutions and the Fermi-Dirac function $f(E, T)$ simulating the experimental thermal broadening. Here, $P = 0.2$ is chosen to account for the particle-hole asymmetric spectral weight observed when tunneling into YSR bands.

The presence of the ETRS along with the particle-hole constraint (symmetry class DIII for $T_{\text{eff}}^2 = -1$ and BDI for $T_{\text{eff}}^2 = 1$) suggests that the system should be described by a different topological invariant than the Majorana number defined for the Kitaev chain with only particle-hole constraint (symmetry class D)[10]. However, the bulk-boundary correspondence cannot be used to conclude on the presence of topologically protected edge modes based on a different classification, because the finite chain typically does not possess an ETRS. The translation along the chain described for the infinite chain above is obviously broken for a finite chain. Odd-length chains have a net magnetic moment; therefore, they cannot be described by any ETRS. Instead of the translation, another lattice symmetry could be combined with the physical time reversal to obtain $T_{\text{eff}}$ for an even-length chain. However, the mirror symmetries proposed in ref. 10 do not hold for the chain built along the $[1\bar{1}1]$ direction. A 180° rotation around the out-of-plane direction at the middle of the chain holds for the considered system, but perturbing one end of the chain breaks the 180° rotation symmetry. Based on the above, only the Majorana number $\mathcal{M}$ introduced for the Kitaev chain can be used as an indication for the presence of topologically non-trivial edge states, since this does not rely on the ETRS. This topological invariant $\mathcal{M}$ can be calculated as:

$$
\mathcal{M} = \text{sgn}\left\{ \text{Pf}\left[\tilde{H}(0)\right] \text{Pf}\left[\tilde{H}(\pi/d)\right] \right\}
$$

(8)

where Pf denotes the Pfaffian and $\tilde{H}(k)$ is the $k$-space Hamiltonian in the Majorana basis[51]. For the present system $\text{Pf}\left[\tilde{H}(k)\right] = -E_0 - t_k$, where $t_k = 2t_1\cos kd + 2t_2\cos 2kd$ is the Fourier transform of the hopping terms. The topological invariant $\mathcal{M}$ takes the value $-1$ ($+1$) for the YSR band crossing the Fermi level an odd (even) number of times between $0$ and $\pi/d$ when the superconducting pairing terms are set to zero. The system is always in the topologically trivial regime in the antiferromagnetic case without SOC ($t_1 = 0$), where $t_k$ is $\pi/d$ periodic and the number of band crossings is even. Changing the parity of the band crossings requires adding strong ETRS-breaking terms to the Hamiltonian. For the extended model described by Eqs. (2), (5) and (6), the boundary of the Brillouin zone is reduced to $\pi/(2d)$ due to the antiferromagnetic structure, and there are two particle-hole pairs for each wave vector. In this case, $\mathcal{M}$ is given by the sign of the product of $\text{Pf}[\tilde{H}(0)] = (-E_0 + 2t_2)^2 + (2\Delta_1)^2$ and $\text{Pf}[\tilde{H}(\pi/(2d))] = (-E_0 - 2t_2)^2 + (2\Delta_1' s_\parallel)^2 - |2h_1 s_\perp|^2$, which simplifies to the same condition as above for $s_\parallel = 0$ and $s_\perp = 1$, and also demonstrates that a finite value of $s_\parallel$ does not prefer the formation of a topologically non-trivial state.

## Data availability

The raw data presented in this work is available on Zenodo (https://doi.org/10.5281/zenodo.7805368).

## Code availability

The code required for data processing and plotting as well as code used for the theoretical simulations is available on Zenodo (https://doi.org/10.5281/zenodo.7805368).

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

## Acknowledgements

We thank Eric Mascot, Roberto Lo Conte and Falko Pientka for helpful discussions. L.S., T.P., J.W., and R.W. gratefully acknowledge funding by the Cluster of Excellence 'Advanced Imaging of Matter' (EXC 2056 - project ID 390715994) of the Deutsche Forschungsgemeinschaft (DFG). L.R. gratefully acknowledges financial support from the National Research, Development and Innovation Office of Hungary via Project Nos. K131938 and FK142601, from the Ministry of Culture and Innovation and the National Research, Development and Innovation Office within the Quantum Information National Laboratory of Hungary (Grant No. 2022-2.1.1-NL-2022-00004) and from the Young Scholar Fund at the University of Konstanz. P.B., J.W. and R.W. acknowledge support by the DFG (SFB 925—project 170620586). R.W. acknowledges funding by the European Union via the ERC Advanced Grant ADMIRE (project No. 786020). T.P. acknowledges support by the DFG (project no. 420120155).

## Author contributions

L.S., P.B., R.W. and J.W. conceived the experiments. L.S. and P.B. performed the measurements and analyzed the experimental data. L.S. performed the numerical simulations in close exchange with T.P. and L.R. L.S. prepared the figures and wrote the manuscript. All authors contributed to the discussions and to correcting the manuscript.

## Funding

## Competing interests

The authors declare no competing interests.
