## [Peer review file · Nature Communications]

REVIEWER COMMENTS

Reviewer #1 (Remarks to the Author):

The manuscript “Testing the topological nature of end states in antiferromagnetic atomic chains on superconductors” investigates the low-energy electronic properties of magnetic chains built using Mn atoms proximitized to two different superconductors (Nb and Ta). The work focuses on elucidating the topological character of the end states.

The data are of high quality, the experimental results are solid and supported by a detailed theoretical model. However, the manuscript does not represent a significant advancement with respect to existing literature which justifies its publication in Nature Communication.

1. In Fig. 1, the authors study Mn chains on Nb. The measurements evidence the occurrence of an end mode at 0.5 meV. In Fig. 3 a, the end mode is perturbed, with the right and left ends now showing slightly different energies, an observation which proves the local nature of the end states, ruling out non-local states like Majoranas. However, the energy of end mode (0.5 meV) already excludes Majoranas.

2. Mn chains on Ta are much more interesting. The data reported in Fig. 2 show a zero-energy mode localized at the chain ends. Fig. 3 d shows that also in this case the mode behaves differently at right and left edges once an additional Mn atom is moved closer to one end (the right end shifts in energy while the left end remains unperturbed). In this case, the approach used by the authors is very useful to analyze the topological nature of the end mode. However, perturbing chains and looking at the response of the end modes to infer their local or non-local character is not a new methodological approach. It has been used by the same group in Nature Nanotechnology 17, 384, where the end modes are perturbed only on one end of the chain by changing the distance between two chains. In this case, the end modes are perturbed only on one end by changing the distance to an additional atom.

3. Antiferromagnetic order has been previously discussed. The experimental data and the theoretical analysis allow to draw similar conclusions to those reported in PRB 104, 045406, which shows how antiferromagnetic order favours large but topologically trivial gaps, whereas ferromagnetic order facilitates the transition in a topologically non-trivial phase.

Reviewer #2 (Remarks to the Author):

In this manuscript, Schneider et al. report on a method for testing the topological nature of end states in adatomic chains on superconducting surfaces. They investigate Mn chains on respectively Nb(110) or Ta(110) surfaces. The latter shows end states at zero bias, but manipulating an extra adatom or a defect near the last adatom in either end is shown not to affect the nearly zero bias peak in the other end, from which it is concluded that this is not a Majorana bound state but rather two separate Fermionic states.

Altogether, this is a very nice piece of work, which reads very well and addresses a timely problem, which should be of interest to most people in the field. It is perhaps stretching it a bit to call this a test of the topological nature, which to many would demand a demonstration of actual braiding, but it is indeed a very direct test of the non-local nature of states exhibiting zero-energy DOS at the ends, which appears to be a necessary condition for the sought for Majorana physics. If the authors can satisfactorily clarify the questions posed below, I would recommend publication.

1) In Fig.1c (Mn@Nb) the left end state remains fixed while the right end state oscillates in energy with chain length. This demonstrates that end states are independent. On the other hand, the finite energy end states observed for Mn@Nb in Fig.3ab both appear to be split by the defect near the left end. This seems to indicate a non-local nature of these states, and therefore I'm confused by the statement at the bottom of p.4, that these two observations both prove the local nature of these states.

2) In Fig.3def, I see an overall shift of all curves which deserves a few words, unless I overlooked those? The sub-gap feature at the left end (3d) is pretty much unaltered, I agree, but it is a little hard to make out how much the sub-gap feature at the left end (3e) is altered. Perhaps an inset with a zoom-in of vertically readjusted near-zero-energy features would make it easier to appreciate this difference?

3) On p.7, the finite-energy end states are said to split for odd N chains. In Fig.1c, however, this happens only in one (the right) end, which was used to argue that they were independent. What exactly is the prediction of the model, should one see the odd-even effect in both ends, and what determines the magnitude of this splitting? The observation of odd-even effect in Fig. 1 is interesting and should perhaps be emphasized a bit more, since it is nicely captured by the model.

4) I find no mentioning of the magnetic anisotropy nor the Dzyaloshinskii-Moriya (DM) interaction for Mn@Nb,Ta. Is it known what is to be expected for these systems, and what is the origin of the AFM ordering, and whether it remain AFM ordered when the substrate is in the normal state? I'm thinking here of the indirect exchange analyzed in your Ref.5, by Schechter et al., where AFM is shown to dominate the phase-diagram in Fig.2a, and the FM phase of the normal state is replaced by a spiral phase with a superconducting substrate. If the ratio between adatom spacing and Fermi wavelength, $a \cdot k_F$, is known in this experiment, it would be nice to know if it is consistent with AFM from this indirect exchange. (Same question could be asked for Ref.23 (Kim et al.), where a spiral ground state was found for Fe@Re, and one might wonder if the $a \cdot k_F$, was simply closer to $(n+1/2)\pi$ in that system, or if it is indeed due to DM-interactions as stated there?). The scenario outlined in Ref.5 is at odds (not in disagreement) with the statement made in the conclusion, that ferromagnetic chains require SOC to become topologically non-trivial, in the sense that with a superconducting substrate, the FM state is in fact unstable towards non-collinear configurations, which support topologically non-trivial phases without any SOC.

Reviewer #3 (Remarks to the Author):

In the past few years, the group of Wiebe and Wiesendanger has done a series of beautiful work on constructing magnetic chains atom-by-atom on clean superconducting substrate and characterizing the magnetic texture and low-energy excitations with the STM. One of the advantages of this approach over the self-assembly grown chains is the ability to design the most desirable geometry and dimension of the magnetic structure. In this work, Schneider et al extend the group's previous work on ferromagnetic chains and fabricate antiferromagnetic (AF) chains of Mn on both superconducting Nb and Ta substrates. They not only measure the spatial dependence of the in-gap Shiba states, but also examine the response of the localized states at the end of the chain to nearby adsorbates or magnetic adatoms. Furthermore, the authors present a minimal model for the antiferromagnetic chain to explain some of the experimental observations.

The study of magnetic chains on superconductors is important because such systems can in principle host Majorana zero modes and this work presents, to my knowledge, the first careful spectroscopic characterization of the AF chains on clean superconducting substrates. The experiment that examines the coupling between single atom and the magnetic chain is also novel. The data quality is very high. Therefore, I view this work as suitable for further consideration in Nature Communication. Below, I provide a list of comments that the authors should address before I recommend publication.

1. My biggest concern comes from the way this paper is packaged and written. At first read, I thought the authors present a new way of distinguishing Majorana modes (MM) from trivial zero edge modes by examining the response of the edge mode to the local defect, as it is suggested by the abstract. However, it is obvious that neither the edge states in Mn chains on Nb or Ta can be interpreted as MM because i) the localized edge states in Mn/Nb appear far away from zero energy (0.51mV!). ii) although the left localized edge state in Mn/Ta resides close to zero energy (but still finite!), the right localized edge state shows double peak at finite energy. Describing and interpreting the experiments in the context of MM and topological superconducting is therefore inappropriate. Because of this, the abstract and part of the main text should be rewritten. Since the spectroscopic data on the chains alone are inconsistent with the MM, the end states are simply trivial without even examining the response to local defect. Hence, the title should be modified as the end states are trivial.

2. Despite the comment above, the work is still interesting and useful for the community. One suggestion for rewriting the paper is to discuss the minimal model as Fig.3 and explain why the AF chains are not in the topological regime, then present the experiment on the local defects on the trivial states and motivate how it may be used to distinguish MM from trivial mode.

3. In line 98, the authors compare the bulk gap $\sim 0.7\text{meV}$ they measured on Mn/Nb to those found in ferromagnetic chains. Such a comparison is misleading as the gap in Mn/Nb is clearly trivial and the gaps in ferromagnetic chains are likely topological. In fact, one can clearly see in Fig. 4f-h, the trivial gap (red shaded region) is generally much larger than the topological gap (blue shaded region) unless the spin-orbit coupling is very large.

High resolution spectroscopic measurements on superconducting AF chains are novel. T

4. What is the nature of the coupling between the local defects and the Shiba chain? In the case of the surface adsorbate (Fig. 3a-b), it seems appropriate to view it as local potential. However, the case of single atom is more complicated because it contains its own energy levels and provides exchange coupling of unknown strength that may or may not change the magnetic texture of the chain. In principle, these two cases can be incorporated in the minimal tight binding model (beyond just shifting E_0) and one can solve it on a large finite size lattice. I think this is very important as it would offer valuable control experiments for future atomic chains system that show zero-bias end states and for “testing the topological nature”.

5. Another motivation for more rigorously integrating the local defects to the tight-binding model is to establish whether or not the spectroscopic properties of the MM on the left end should change if the right MM is perturbed by a local defect.

6. Minor point, the ordering of the panels in fig4 is a bit chaotic.

Overall, this work presents an interesting experiment with high quality data, but the interpretation and claims needs be revised.

Yellow marked text = here we added or changed something in the manuscript/supplement
The corresponding changes appear as **red text** in the manuscript/supplement

Reviewer #1 (Comments for the Author):

The manuscript “Testing the topological nature of end states in antiferromagnetic atomic chains on superconductors” investigates the low-energy electronic properties of magnetic chains built using Mn atoms proximitized to two different superconductors (Nb and Ta). The work focuses on elucidating the topological character of the end states.

The data are of high quality, the experimental results are solid and supported by a detailed theoretical model. However, the manuscript does not represent a significant advancement with respect to existing literature which justifies its publication in Nature Communication.

We thank the Reviewer for the appreciation of our work’s quality. In the following, we will clarify why we strongly disagree with the Reviewer’s further claim and why we are confident that it *does* constitute a significant advancement in the understanding of magnet-superconductor hybrids, thus justifying publication in Nature Communications. In short, there is no publication so far that 1) explains on a minimal model level the ubiquity of trivial finite energy end states in AFM chains, and 2) experimentally tests a method to prove the trivial nature of such end states. The statements in the second paragraph of Reviewer #3’s report also reassure this.

1. In Fig. 1, the authors study Mn chains on Nb. The measurements evidence the occurrence of an end mode at 0.5 meV. In Fig. 3 a, the end mode is perturbed, with the right and left ends now showing slightly different energies, an observation which proves the local nature of the end states, ruling out non-local states like Majoranas. However, the energy of end mode (0.5 meV) already excludes Majoranas.

While this is indeed true for semi-infinite systems, it has been shown in previous experiments that the (precursor) Majorana modes in finite-size systems can hybridize and shift away from E_F (e.g. Ref. 26). They may even remain at finite energies for large system sizes due to fine-tuning of their interaction (see e.g. Mier *et al.*, Phys. Rev. Research **4**, L032010 (2022)). Therefore, the off-zero energy end states may still be such precursors of Majorana modes which evolve into localized Majorana modes for sufficiently long chains. Furthermore, we do not imply in the manuscript that the end states in the chain on the Nb substrate are Majorana modes, but even if they are finite-energy modes, they could be other electronic modes of topological origin and as such stable against perturbations. Using our perturbative method, we can experimentally demonstrate that this is not the case.

Nevertheless, we agree that the Mn chain on Ta(110) represents the more interesting case in terms of Majorana physics since here, the modes are near-zero energy and the spin-orbit coupling is expected to be higher compared to Nb. However, the preceding discussion of Mn chains on Nb(110) is required to gain a more substantial understanding of the trivial end states

first, which is a central part of our manuscript. We believe that it is supporting a canonical introduction to the presented method to start with a case where - due to our enhanced effective energy resolution on the Nb substrate compared to Ta – we can assign sub-gap features such as end states and quasiparticle interference phenomena much more clearly. Hence, we can discuss the emergence of this new type of end state and compare it with the theoretical model in Fig. 4. We believe that this observation and characterization of a novel type of localized state is interesting on its own and that is why we designated a significant portion of the manuscript to this discussion. After establishing this, as the Reviewer admits, Fig. 2 still includes a useful showcase for the Ta system where we apply our perturbative method.

2. Mn chains on Ta are much more interesting. The data reported in Fig. 2 show a zero-energy mode localized at the chain ends. Fig. 3 d shows that also in this case the mode behaves differently at right and left edges once an additional Mn atom is moved closer to one end (the right end shifts in energy while the left end remains unperturbed). In this case, the approach used by the authors is very useful to analyze the topological nature of the end mode. However, perturbing chains and looking at the response of the end modes to infer their local or non-local character is not a new methodological approach. It has been used by the same group in Nature Nanotechnology 17, 384, where the end modes are perturbed only on one end of the chain by changing the distance between two chains. In this case, the end modes are perturbed only on one end by changing the distance to an additional atom.

We note that the Reviewer acknowledges the appropriateness of our experimental method. However, there has to be a misunderstanding, as it is clearly not true that the behavior found in our previous work (Ref. 26) is similar to the one found here. In fact, the precursors of Majorana modes observed in Ref. 26 are found to change on *both* ends simultaneously when only one of the ends is perturbed, thus demonstrating their *non-local* nature. This effect is based on the hybridization of two precursor Majorana modes that reside at the ends of two equivalent chains (c.f. Fig. 5 of Ref. 26). The hybridization, *per se*, is no proof of the topological nature of the modes. Two chains with equal end modes of equal type would let the end modes hybridize irrespective of being topologically trivial or nontrivial (Majorana modes hybridize with Majorana modes, Dirac-fermionic end states hybridize with Dirac-fermionic ones). The new aspect of the current manuscript is to perturb an end mode with a perturbation that would leave a topologically nontrivial state at zero energy and merely laterally shift it, and would lift the topologically trivial states, thereby distinguishing between both cases. In the submitted manuscript, we test this method for a situation where it is not clear whether the end states are of trivial or topological character. In stark contrast to the findings in Ref. 26, the states in the present manuscript are found to be clearly *local*, as the Reviewer has summarized accurately. Thus, the outcome of this test is just opposite to the one in Ref. 26 and constitutes a novel finding. This finally proves that this perturbative method works in order to exclude the topological case. To clarify this, **we have revised the title and the abstract of our manuscript.**

3. Antiferromagnetic order has been previously discussed. The experimental data and the theoretical analysis allow to draw similar conclusions to those reported in PRB 104, 045406, which shows how antiferromagnetic order favours large but topologically trivial gaps, whereas ferromagnetic order facilitates the transition in a topologically non-trivial phase.

We agree that antiferromagnetic order has been discussed in other publications before, which has not been sufficiently acknowledged in the originally submitted form of the manuscript. Thus, already prior to the Reviewer's reports, we have realized this ourselves and slightly changed a sentence in the introduction of our manuscript (page 2, line 71). However, we are convinced that no data of any comparable quality has been published together with an interpretation linking the magnetic texture of the nanostructures to in-gap features. Firstly, the study of Mier *et al.* mentioned by the Reviewer does not show spin-polarized data on the magnetic structure. Thus, the assumed magnetic ground state of the chain is not directly confirmed by experimental data. Secondly, they find that the sub-gap states do not hybridize sufficiently in order to form bands of any significant bandwidth. Therefore, the chains presented in Fig. 2 of Mier *et al.* realize a different regime, comparable to an "atomically insulating" state of YSR resonances, which is unquestionably topologically trivial. In our case, we observe quasiparticle interference features inside the gap, thereby proving that the YSR states form bands of significant width where a topologically non-trivial phase may be formed. The gaps we find experimentally are rather caused by a gap reopening (c.f. Fig. 4b-d of the manuscript) and the chains are thus in a completely different phase than the ones studied by Mier *et al.*. Thirdly, Mier *et al.* speculate on the magnetic ground state of the chain in Fig. 3 of the reference to be a 120° spin spiral instead of an antiferromagnetic state. The bottom line of the work by Mier *et al.* is that the YSR bands in antiferromagnetically ordered chains are not expected to be dispersing (page 6, line 6 of Mier *et al.*), which is in clear contrast to our observation.

In fact, we did not find the statement on the nature of the gaps depending on the magnetic texture made by the Reviewer in the Mier *et al.* paper. Rather, the calculations in Fig. 3 look as if the antiferromagnetic state was more or less gapless (this is hard to judge due to the small number of atoms in the chain leading to strong finite-size quantization) while spin helical ground states provide large but topologically trivial gaps. The theoretical prediction of large but topologically trivial gaps being common in antiferromagnetic chains, together with their explanation based on time-reversal-symmetry breaking and by taking spin-orbit coupling into account, is thus a novel result of our work, which does not appear to have been discussed in the literature before. For completeness, we have added the work by Mier *et al.* as Ref. 34 to our manuscript.

Another work studying the sub-gap structure of antiferromagnetic YSR chains is the very recent one by Küster and coworkers (Ref. 33 of our manuscript). For their samples, the antiferromagnetic texture of one of the chains has been experimentally demonstrated using spin-polarized STM data presented in another publication (Brinker *et al.*, Sci. Adv. **8**, abi7291 (2022)). Nonetheless, they do not discuss the implications of the antiferromagnetic texture on the sub-gap states at all. The focus of this work is rather on the ubiquitous appearance of end

states in all of their chains, regardless of the magnetic texture. Motivated by our finding that finite-energy end states are ever-present in our minimal model for antiferromagnetic chains shown in Fig. 4, we have revisited the data of Ref. 33. Interestingly, in Fig. 2B and 2D of Ref. 33, very similar, strongly-localized end states are found on exactly the chains with antiferromagnetic order. This provides additional evidence that our findings are generalizable to lots of other magnet-superconductor hybrid systems and we have added two sentences to our manuscript (page 2, line 71 and page 7, lines 310ff.) highlighting this exciting result concerning antiferromagnetic chains.

After clarifying these points, we hope that we were able to convince the Reviewer that our experiments constitute a significant advantage over previous data reported in literature. Since apart from this, the Reviewer has undoubtedly praised the quality of our work, we are confident that it qualifies for publication in Nature Communications now.

Reviewer #2 (Comments for the Author):

In this manuscript, Schneider et al. report on a method for testing the topological nature of end states in atomic chains on superconducting surfaces. They investigate Mn chains on respectively Nb(110) or Ta(110) surfaces. The latter shows end states at zero bias, but manipulating an extra adatom or a defect near the last adatom in either end is shown not to affect the nearly zero bias peak in the other end, from which it is concluded that this is not a Majorana bound state but rather two separate Fermionic states.

Altogether, this is a very nice piece of work, which reads very well and addresses a timely problem, which should be of interest to most people in the field. It is perhaps stretching it a bit to call this a test of the topological nature, which to many would demand a demonstration of actual braiding, but it is indeed a very direct test of the non-local nature of states exhibiting zero-energy DOS at the ends, which appears to be a necessary condition for the sought for Majorana physics. If the authors can satisfactorily clarify the questions posed below, I would recommend publication.

We thank the Reviewer for valuing the quality of our work and agree that our claims must not be stretched too far. Thus, to adequately highlight the actual qualities of the work, we have changed the title of the manuscript to “Probing the topologically trivial nature of end states in antiferromagnetic atomic chains on superconductors”. Moreover, we have revised the corresponding passages in the abstract (see also the reply to Reviewer #3).

1) In Fig.1c (Mn@Nb) the left end state remains fixed while the right end state oscillates in energy with chain length. This demonstrates that end states are independent. On the other hand, the finite energy end states observed for Mn@Nb in Fig.3ab both appear to be split by the defect near the left end. This seems to indicate a non-local nature of these states, and therefore I'm confused by the statement at the bottom of p.4, that these two observations both prove the local nature of these states.

We thank the Reviewer for calling our attention to this slight inaccuracy. As one can see in Figs. 3a,b, the defect on the left side of the chain is not the only one that got removed by bias pulsing in this field of view, but it is the most clearly visible one. Closely comparing the topographies in Fig. 3a and 3b shows another defect on the lower side of the chain and yet another one on the top right part of the image, both of which disappeared after pulsing. Therefore, to be more accurate, both sides were perturbed in Fig. 3a, but not by the same amount. However, when comparing the individual spectra of both sides "before" and "after" pulsing, it is apparent that the right side's end state did not change severely in energy whereas the left one did. That is why we did not focus on the other defects in the previous version of the manuscript. However, we agree that this may be misleading. Thus, we have slightly changed the selection of dI/dV maps in Figs. 3a and b to avoid confusion and we provide additional information in the new Supplementary Note 4 discussing the role of additional mobile defects in Figs. 3a,b. Moreover, we show individual spectra from this experiment in a new Supplementary Figure 4 in order to clarify this point.

2) In Fig.3def, I see an overall shift of all curves which deserves a few words, unless I overlooked those? The sub-gap feature at the left end (3d) is pretty much unaltered, I agree, but it is a little hard to make out how much the sub-gap feature at the left end (3e) is altered. Perhaps an inset with a zoom-in of vertically readjusted near-zero-energy features would make it easier to appreciate this difference?

It is true that the differences are hardly visible in the previous representation, we thank the Reviewer for this suggestion and changed the figure accordingly. In fact, in the caption of Figs. 3d,e,f, we do state that "The spectra are vertically offset for clarity", which the Reviewer probably overlooked. Moreover, we now present additional data with the full line-profiles in the new Supplementary Figure 5 to make the effect clearer.

3) On p.7, the finite-energy end states are said to split for odd N chains. In Fig.1c, however, this happens only in one (the right) end, which was used to argue that they were independent. What exactly is the prediction of the model, should one see the odd-even effect in both ends, and what determines the magnitude of this splitting? The observation of odd-even effect in Fig. 1 is interesting and should perhaps be emphasized a bit more, since it is nicely captured by the model.

We thank the Reviewer for emphasizing the importance of this effect. We have slightly altered a sentence on page 7, lines 279 and 280 to highlight the agreement between theory and experiment.

In fact, the even-odd effect is theoretically expected and experimentally observed on both sides. Notably, what we refer to is the length regime of $8 < N < 14$ where a clear even-odd effect can be seen in the data of Fig. 1c. The other oscillatory change in energy mentioned by the Reviewer takes place at $15 < N < 20$ and can be related to local defects since it only appears on one of the sides. Thus, it is correct that this is a hallmark of a local state. The fact that this seems to follow an even-odd-like pattern as well may just be a coincidental feature.

The model predicts a twofold degeneracy protected by effective time reversal symmetry of all states for even chain lengths - but this does not apply for odd ones. However, it is not *a priori* clear how large the degeneracy lifting will be for odd chain lengths, just that it can be finite. Therefore, the magnitude of the splitting changes strongly with different model parameters. Nevertheless, the period of the oscillatory splitting is always given to be $\Delta N = 2$ by the symmetry arguments mentioned above.

4) I find no mentioning of the magnetic anisotropy nor the Dzyaloshinskii-Moriya (DM) interaction for Mn@Nb,Ta. Is it known what is to be expected for these systems, and what is the origin of the AFM ordering, and whether it remain AFM ordered when the substrate is in the normal state? I'm thinking here of the indirect exchange analyzed in your Ref.5, by Schechter et al., where AFM is shown to dominate the phase-diagram in Fig.2a, and the FM phase of the normal state is replaced by a spiral phase with a superconducting substrate. If the ratio between adatom spacing and Fermi wavelength, a^*k_F , is known in this experiment, it would be nice to know if it is consistent with AFM from this indirect exchange. (Same question could be asked for Ref.23 (Kim et al.), where a spiral ground state was found for Fe@Re, and one might wonder if the a^*k_F , was simply closer to $(n+1/2)\pi$ in that system, or if it is indeed due to DM-interactions as stated there?). The scenario outlined in Ref.5 is at odds (not in disagreement) with the statement made in the conclusion, that ferromagnetic chains require SOC to become topologically non-trivial, in the sense that with a superconducting substrate, the FM state is in fact unstable towards non-collinear configurations, which support topologically non-trivial phases without any SOC.

Experimentally, we have verified that magnetic contrast compatible with antiferromagnetic order is observed both below and above the critical field H_c for chains on Nb and Ta. Accordingly, the antiferromagnetic phase discussed in this manuscript is not altered by the presence of

superconductivity but robust. While the Reviewer's suggestion to compare the ratio of adatom spacing and the Fermi wavelength is indeed very interesting, it is only meaningful in the case of RKKY mediated interactions. The distance between Mn adatoms in the structures presented here is small, such that direct overlap of the d -orbitals presumably dominates the magnetic interactions.

Theoretically, the magnetic interactions and anisotropies for Mn on Nb(110) have been calculated by mapping *ab-initio* results onto a generalized Heisenberg model. The results are published in Ref. 41. The dimers and chains along the $[1\bar{1}1]$ direction exhibit a strong isotropic nearest-neighbor antiferromagnetic Heisenberg exchange coupling of more than 30 meV, whereas the DM interaction is almost two orders of magnitude smaller. The former fact points towards the role of strong and direct d -orbital overlap as speculated above while the latter can be attributed to the relatively small SOC in Nb. The magnetic anisotropy energy per atom preferring the out-of-plane direction was found to be around 0.5 meV, comparable in magnitude to the DM interaction. It was concluded that the strong nearest-neighbor Heisenberg coupling together with the anisotropy stabilizes the antiferromagnetic order over the spin spiral preferred by the weak DMI. These results are in good agreement with spin-polarized data of, e.g., Ref. 37.

Ab-initio calculations on Mn/Ta(110) are still ongoing and are planned to be used for a future publication, but strong similarities between Nb and Ta (see e.g. Refs. 28 & 38) point towards a similar mechanism stabilizing the magnetic order. Preliminary results yield a similar isotropic Heisenberg exchange but a larger DM interaction compared to Mn/Nb(110). Nevertheless, the DM interaction is not sufficient to drive the chain into a non-collinear ground state.

Instability towards a spiral under SC is indeed an interesting additional mechanism to create topological SC in FM chains. However, it does not always appear since measurements showing the FM state below the transition temperature have been reported by various groups on different platforms (see, e.g., Ref. 24, 26 or 37). The theoretical model presented by *Schechter et al.* predicts that a collinear state transforms into a spin spiral state with wave vector $q \approx \xi^{-1}$, where ξ is the coherence length. Since the coherence length is at least 10 nm, the period of the spin spiral would be above 40 atoms, which is at the upper limit of the chain lengths investigated in this work. Furthermore, the energy gain per spin from forming the spin spiral state is $\Delta E \approx Jq^2d^2$, where J is the Heisenberg exchange and d is the chain spacing. Using the Heisenberg exchange obtained from *ab-initio* calculations (30 meV) for the Nb substrate and the period estimated above, this results in an energy gain on the order of 10 μ eV, which is one order of magnitude smaller than the anisotropy energy (0.5 meV) stabilizing the collinear order. This indicates that the effect on superconductivity on the magnetic structure in closely packed chains is much weaker than the influence of spin-orbit coupling on the spin model parameters (anisotropy, DMI), although the latter were not considered in Ref. 5. Recent model calculations taking spin-orbit coupling into account corroborate the fact that small superconducting pairing terms have typically little influence on collinear magnetic phases (c.f. Neuhaus-Steinmetz *et al.*, Phys. Rev. B **105**, 165415 (2022)). In this work, only a number of very special phases with very low exchange interaction energy were found that could be affected (for instance a 3 up, 3 down phase).

Reviewer #3 (Comments for the Author):

In the past few years, the group of Wiebe and Wiesendanger has done a series of beautiful work on constructing magnetic chains atom-by-atom on clean superconducting substrate and characterizing the magnetic texture and low-energy excitations with the STM. One of the advantages of this approach over the self-assembly grown chains is the ability to design the most desirable geometry and dimension of the magnetic structure. In this work, Schneider et al extend the group's previous work on ferromagnetic chains and fabricate antiferromagnetic (AF) chains of Mn on both superconducting Nb and Ta substrates. They not only measure the spatial dependence of the in-gap Shiba states, but also examine the response of the localized states at the end of the chain to nearby adsorbates or magnetic adatoms. Furthermore, the authors present a minimal model for the antiferromagnetic chain to explain some of the experimental observations.

The study of magnetic chains on superconductors is important because such systems can in principle host Majorana zero modes and this work presents, to my knowledge, the first careful spectroscopic characterization of the AF chains on clean superconducting substrates. The experiment that examines the coupling between single atom and the magnetic chain is also novel. The data quality is very high. Therefore, I view this work as suitable for further consideration in Nature Communication. Below, I provide a list of comments that the authors should address before I recommend publication.

We thank the Reviewer for this very positive feedback and acknowledgement that our work contains two main novel points justifying publication in Nature Communications.

1. My biggest concern comes from the way this paper is packaged and written. At first read, I thought the authors present a new way of distinguishing Majorana modes (MM) from trivial zero edge modes by examining the response of the edge mode to the local defect, as it is suggested by the abstract. However, it is obvious that neither the edge states in Mn chains on Nb or Ta can be interpreted as MM because i) the localized edge states in Mn/Nb appear far away from zero energy (0.51meV!). ii) although the left localized edge state in Mn/Ta resides close to zero energy (but still finite!), the right localized edge state shows double peak at finite energy. Describing and interpreting the experiments in the context of MM and topological superconducting is therefore inappropriate. Because of this, the abstract and part of the main text should be rewritten. Since the spectroscopic data on the chains alone are inconsistent with the MM, the end states are simply trivial without even examining the response to local defect. Hence, the title should be modified as the end states are trivial.

We thank the Reviewer for these suggestions. However, we do not agree with the statement that it would be immediately clear that the observed end states in the two systems are topologically trivial. First, concerning the Mn/Nb system, we have shown in previous experiments, that the precursor Majorana modes in finite-size systems can hybridize and shift away from E_F (Ref. 26, see also the reply to Reviewer #1). Therefore, merely an off-zero energy of

an end state alone is not a sufficient criterion to conclude on the topologically trivial and local nature of those end modes. Second, concerning the Mn/Ta system: In the originally submitted version of the manuscript, it was indeed already visible in Fig. 2b that the two end states in the Mn₂₂ chain have slightly different energy, thus anticipating the result of Figs. 3c-f. This splitting likely results from additional close-by impurities, yet, we chose the dataset of that Mn₂₂ chain to be consistent with the chain length in Fig. 2a. We agree with Reviewer #3 that, thereby, it was in fact confusing for the reader why the experiment in Figs. 3c-f is required to judge on the topological or trivial nature of the end states in Mn/Ta. For this reason, in the revised manuscript, we have replaced the line-profile in Fig. 2b with another line-profile of a Mn₂₀ chain measured with the same tip and at the same position on the sample like the previous panel. Notably, in this panel, both end states are not split and at zero-energy within our experimental energy resolution. We emphasize at this point that other candidate platforms for topological boundary modes often lack some degree of tunability – e.g. the option to change the exact number of atoms in an atomic chain. Thus, from such a dataset on the Mn₂₀ chain alone, we were unable to tell whether the end states are topological Majorana modes or trivial local features and additional testing schemes are required. We believe that the overall storyline of the work is improved by showing this new dataset instead of the Mn₂₂ data. The previous Fig. 2b has now been shifted to Supplementary Figure 5.

However, we also agree that it would be appropriate to adopt the Reviewers advice and renamed the title to “Probing the topologically trivial nature of end states in antiferromagnetic atomic chains on superconductors”. Moreover, we have revisited critical parts in the abstract.

2. Despite the comment above, the work is still interesting and useful for the community. One suggestion for rewriting the paper is to discuss the minimal model as Fig.3 and explain why the AF chains are not in the topological regime, then present the experiment on the local defects on the trivial states and motivate how it may be used to distinguish MM from trivial mode.

Please see our comments to point 1. We thank the Reviewer for the suggestion, but we believe that the order of figures is still appropriate. The way the manuscript is written right now, we first present the experimental findings and characterization of the trivial end states and add our model calculations at a later point to explain the results in a theoretical framework.

3. In line 98, the authors compare the bulk gap $\sim 0.7\text{meV}$ they measured on Mn/Nb to those found in ferromagnetic chains. Such a comparison is misleading as the gap in Mn/Nb is clearly trivial and the gaps in ferromagnetic chains are likely topological. In fact, one can clearly see in Fig. 4f-h, the trivial gap (red shaded region) is generally much larger than the topological gap (blue shaded region) unless the spin-orbit coupling is very large.

We thank the Reviewer for this suggestion. However, we would like to point out that at this point in the manuscript it is not obvious that the gap is topologically trivial. This would only be the

case if the chain would be in the atomically insulating limit (cf. our reply to the third comment of Reviewer #1), but the hybridization of the YSR states is clearly visible in the data which would allow for the bands crossing the Fermi level and the possibility for a topologically non-trivial gap. In the sentence mentioned by the Reviewer, however, we do not make any claims on the trivial or topological nature of the gaps, but only comment on the gap sizes. Only following a theoretical modelling can we conclude that even if the YSR bands cross the Fermi level, the antiferromagnetic structure opens a large gap, which, however, is likely topologically trivial. We rather openly discuss the fact that “unrealistically large values of SOC have to be assumed in order to enable the formation of MMs” in AFM chains (page 7, line 302). Thus, we think that our statements are correct and not misleading.

4. What is the nature of the coupling between the local defects and the Shiba chain? In the case of the surface adsorbate (Fig. 3a-b), it seems appropriate to view it as local potential. However, the case of single atom is more complicated because it contains its own energy levels and provides exchange coupling of unknown strength that may or may not change the magnetic texture of the chain. In principle, these two cases can be incorporated in the minimal tight binding model (beyond just shifting E_0) and one can solve it on a large finite size lattice. I think this is very important as it would offer valuable control experiments for future atomic chains system that show zero-bias end states and for “testing the topological nature”.

We agree with the Reviewer that these simulations would be helpful. Therefore, we have added the new panels e-h to Supplementary Figure 1 and an accompanying discussion to Supplementary Note 1 showing the influence of an additional lattice site locally coupled to one of the chain ends for the three different cases. The results show qualitatively similar behavior to the ones where E_0 is varied. Namely, while the precursors of Majorana modes which reside in the small-gap topologically non-trivial phase split into finite-energy states on *both* chain ends simultaneously although only the right side is perturbed, the Majorana modes (MMs) residing in the large-gap topologically non-trivial phase do not split but merely laterally shift. In contrast, the trivial end state on the perturbed side of the chain can be individually split in energy while the end state on the other side stays unaffected, thereby proving their local nature and validating our approach.

Concerning the coupling mechanism of the adatom to the chain: since the chain atoms are densely packed, we expect their exchange coupling to be very large (c.f. Ref. 41) and the exchange coupling provided by the additional adatom to be just a small perturbation since the magnetic interactions have a relatively short range. Both the experiments and the *ab-initio* calculations indicate that the antiferromagnetic ground state is robust against changing the chain length, meaning that a single additional atom is probably not strong enough to alter the magnetic ground state. Therefore, we would conclude that hybridization between the adatom’s energy levels and the chain’s energy levels are the most relevant contribution. In particular, direct *d*-orbital overlap is weak for the distances between the adatom and the terminal chain atom compared to the *d*-orbital overlap of atoms *within* the chain. In order to interpret the sub-gap

spectrum of the ensemble, we expect hybridizations between via YSR states of the atom and the chain to be the dominant factor for changes in the sub-gap electronic structure.

We note at this point that a local change in the magnetic texture at one of the ends is not expected to change the bulk topology of the chain. Therefore, in the large-gap topological phase, the Majorana mode will not be destroyed but only shift laterally (c.f. Ref. 35). We have added more data to the new Supplementary Note 5 in order to rule out this effect for the perturbed Mn chains on Ta(110).

5. Another motivation for more rigorously integrating the local defects to the tight-binding model is to establish whether or not the spectroscopic properties of the MM on the left end should change if the right MM is perturbed by a local defect.

We thank the Reviewer for this suggestion and have added such a simulation to Supplementary Figure 1 and its description to Supplementary Note 1. We find the following: Majorana modes do not split and precursors of Majorana modes split simultaneously on both ends, in agreement with their non-local nature and previous studies.

6. Minor point, the ordering of the panels in fig4 is a bit chaotic.

We have done a few minor rearrangements in Figure 4 to have a clearer spacing between the panels and an arrangement of the panels according to their appearance in the text.

Overall, this work presents an interesting experiment with high quality data, but the interpretation and claims needs be revised.

We thank the Reviewer for all the helpful suggestions and hope that with all the according revisions, our interpretations and claims are now very clear.

REVIEWERS' COMMENTS

Reviewer #1 (Remarks to the Author):

I read with interest the reply of the authors to my comments. I appreciate their detailed answers and explanations. As already mentioned in my first report, I find the work very solid and the quality of the data impressive. However, I raised concerns about the advancement of the present work with respect to existing literature. The authors addressed these points in their revision and I find their response convincing. Before publication, I would ask the authors to include some of their explanations directly in the manuscript, considering the following comments.

1. Experimental data presented in Fig. 1

On the “obviously” trivial origin of the end states. The authors base the relevance of these data on the fact that end modes “may even remain at finite energies for large system sizes due to finite tuning of their interaction”. I encourage the authors to mention this aspect in the manuscript and to reference to Phys. Rev. Research 4, L032010.

2. Methodological approach based on perturbing one end.

In my comment, I referred to the methodological approach: coupling the end states to a perturbation (either an additional chain or an additional atom), not to the outcome (local or non-local). In any case, I very much like the additional panels and related discussion in Supplementary Note 1 as well as the data reported in Supplementary Figure 5. They help stressing the behavior expected for topological states compared to local states.

3. Antiferromagnetic chains.

Comments on Ref. 34: without spin-orbit interaction, antiferromagnetically coupled YSR states are localized on the atom, preventing the realization of dispersive bands and resulting in a trivial regime. In this work, without spin orbit interaction ($t \rightarrow 0$ in the model) the hopping between antiferromagnetically coupled nearest neighbor atoms is suppressed, also resulting in a trivial regime. The calculations reported in Ref. 34 (Fig. 3b) show the appearance of a zero energy end state, whose intensity decays inside the bulk. Here, when the spin-orbit interaction is switched on, the system might enter into a topologically non trivial regime and there are zero energy end states rapidly decaying inside the bulk. The bands are clearly dispersing, but it would be appropriate citing these previous studies (Ref. 33 and PRL PRL 120,167001) when discussing the antiferromagnetic results. The in-gap bands are experimentally clearly visible for Nb while for Ta (the most interesting case) the visualization of in-gap dispersing bands is more subtle.

There is an important aspect mentioned in the reply, which I sincerely did not catch in the first version of the manuscript: “The gaps we find experimentally are rather caused by a gap reopening”. Given the importance of gap reopening for the possible emergence of topological properties, I think this should be clearly mentioned in the manuscript.

Comments on Ref. 33: as mentioned by the authors, the role of the magnetic texture is not discussed at all.

Reviewer #2 (Remarks to the Author):

I am perfectly happy with the detailed response given to my questions and recommend this manuscript for publication.

Reviewer #3 (Remarks to the Author):

In the rebuttal letter, the authors have adequately addressed most of my and the other reviewers' comments. I am still not satisfied with their reply to my comment 1, but the revised title and abstract are now much more appropriate. Since the experimental concept is valid and the data quality is very high, I think it would be a valuable addition for the community. Therefore, I recommend publication.

Yellow marked text = here we added or changed something in the manuscript/supplement
The corresponding changes appear as red text in the manuscript/supplement

Reviewer #1 (Remarks to the Author):

I read with interest the reply of the authors to my comments. I appreciate their detailed answers and explanations. As already mentioned in my first report, I find the work very solid and the quality of the data impressive. However, I raised concerns about the advancement of the present work with respect to existing literature. The authors addressed these points in their revision and I find their response convincing. Before publication, I would ask the authors to include some of their explanations directly in the manuscript, considering the following comments.

We thank the Reviewer for appreciating the changes and for recommending publication.

1. Experimental data presented in Fig. 1

On the “obviously” trivial origin of the end states. The authors base the relevance of these data on the fact that end modes “may even remain at finite energies for large system sizes due to finite tuning of their interaction”. I encourage the authors to mention this aspect in the manuscript and to reference to Phys. Rev. Research 4, L032010.

We thank the Reviewer for this suggestion, which we have implemented on page 3, lines 152-155.

2. Methodological approach based on perturbing one end.

In my comment, I referred to the methodological approach: coupling the end states to a perturbation (either an additional chain or an additional atom), not to the outcome (local or non-local). In any case, I very much like the additional panels and related discussion in Supplementary Note 1 as well as the data reported in Supplementary Figure 5. They help stressing the behavior expected for topological states compared to local states.

We thank the Reviewer for acknowledging our additional data in the Supplementary Information Figures.

3. Antiferromagnetic chains.

Comments on Ref. 34: without spin-orbit interaction, antiferromagnetically coupled YSR states are localized on the atom, preventing the realization of dispersive bands and resulting in a trivial regime. In this work, without spin orbit interaction ($t_1 \rightarrow 0$ in the model) the hopping between antiferromagnetically coupled nearest neighbor atoms is suppressed, also resulting in a trivial regime. The calculations reported in Ref. 34 (Fig. 3b) show the appearance of a zero energy end state, whose intensity decays inside the bulk. Here, when the spin-orbit interaction is switched on, the system might enter into a topologically non trivial regime and there are zero energy end states rapidly decaying inside the bulk. The bands are clearly dispersing, but it would be appropriate citing these previous studies (Ref. 33 and PRL PRL 120,167001) when discussing the antiferromagnetic results. The in-gap bands are experimentally clearly visible for Nb while for Ta (the most interesting case) the visualization of in-gap dispersing bands is more subtle.

There is an important aspect mentioned in the reply, which I sincerely did not catch in the first version of the manuscript: “The gaps we find experimentally are rather caused by a gap reopening”. Given the importance of gap reopening for the possible emergence of topological properties, I think this should be clearly mentioned in the manuscript.

Comments on Ref. 33: as mentioned by the authors, the role of the magnetic texture is not discussed

at all.

We thank the Reviewer for this suggestion and added a sentence on the gap reopening on page 4, line 218-219. We have further added two references (PRL 120, 167001 (2018) as suggested by the Reviewer and the simultaneously published PRL 120, 156803 (2018)) on antiferromagnetic YSR dimers on page 1, line 53 and we now cite the paper by Küster et al. again on page 2, line 98.

Reviewer #2 (Remarks to the Author):

I am perfectly happy with the detailed response given to my questions and recommend this manuscript for publication.

We thank the Reviewer for the kind appreciation of our reply and for the recommendation to publish our work.

Reviewer #3 (Remarks to the Author):

In the rebuttal letter, the authors have adequately addressed most of my and the other reviewers' comments. I am still not satisfied with their reply to my comment 1, but the revised title and abstract are now much more appropriate. Since the experimental concept is valid and the data quality is very high, I think it would be a valuable addition for the community. Therefore, I recommend publication.

We thank the Reviewer for the recommendation to publish our manuscript.